# Gene Abnormalities and Modulated Gene Expression Associated with Radionuclide Treatment: Towards Predictive Biomarkers of Response

**DOI:** 10.3390/genes15060688

**Published:** 2024-05-26

**Authors:** Tim A. D. Smith

**Affiliations:** Nuclear Futures Institute, School of Computer Science and Engineering, Bangor University, Dean Street, Bangor LL57 1UT, UK; tim.smith@bangor.ac.uk

**Keywords:** molecular radiotherapy, radionuclides, cancer, biomarker, predictive

## Abstract

Molecular radiotherapy (MRT), also known as radioimmunotherapy or targeted radiotherapy, is the delivery of radionuclides to tumours by targeting receptors overexpressed on the cancer cell. Currently it is used in the treatment of a few cancer types including lymphoma, neuroendocrine, and prostate cancer. Recently reported outcomes demonstrating improvements in patient survival have led to an upsurge in interest in MRT particularly for the treatment of prostate cancer. Unfortunately, between 30% and 40% of patients do not respond. Further normal tissue exposure, especially kidney and salivary gland due to receptor expression, result in toxicity, including dry mouth. Predictive biomarkers to select patients who will benefit from MRT are crucial. Whilst pre-treatment imaging with imaging versions of the therapeutic agents is useful in demonstrating tumour binding and potentially organ toxicity, they do not necessarily predict patient benefit, which is dependent on tumour radiosensitivity. Transcript-based biomarkers have proven useful in tailoring external beam radiotherapy and adjuvant treatment. However, few studies have attempted to derive signatures for MRT response prediction. Here, transcriptomic studies that have identified genes associated with clinical radionuclide exposure have been reviewed. These studies will provide potential features for seeding multi-component biomarkers of MRT response.

## 1. Introduction

### 1.1. Radionuclide Therapy and MRT

External beam radiotherapy (EBRT) is a highly effective cancer treatment but is generally localised to primary tumours and local disease. Radionuclide therapy includes molecular radiotherapy (MRT), also called targeted radiotherapy or radioimmunotherapy, which is administered systemically and will target the primary tumour and local and distant spread. MRT is delivered via the antibodies or ligands of cell surface receptors upregulated on cancer cells conjugated with cytotoxic radionuclides. Whilst EBRT is used in the treatment of about 50% of cancers, MRT is currently used in treating just a few indications, including lymphoma, neuroendocrine and prostate tumours, characterised by a cancer-specific cell surface expression of the receptors CD20, somatostatin and PSMA, respectively. Figure 1 shows a historical clinical perspective of MRT. The outcome from recent MRT trials in prostate cancer (VISION) and neuroendocrine (NETTER) have been practice-changing, demonstrating significantly improved progression free- (PFS) and overall survival (OS) [1]. These findings have increased interest in applying MRT to more prostate cancer patients and more broadly to other tumour types. However, 30–40% of patients treated with MRT continue to show progression [2]. Robust markers of response to MRT are required to ensure an informed approach to patient selection.

The radiation dose delivered during an MRT fraction is dependent on the delivery of the MRT agent to the cancer cells and the cell surface target expression, both of which will be tumour specific. Serial imaging to measure the amount and duration of radionuclide binding to a tumour is essential to ensure patient suitability for MRT and to tailor dose [3]. Like EBRT, the response of the tumour is dependent on the cancer cell’s intrinsic radiosensitivity and microenvironmental factors, particularly O_2_ level [4]. Hypoxic tumours are known to be radioresistant and sensitive to a radiotherapy fractionation regimen, which can be resolved by hypoxia modification [5,6]. Gene expression-based biomarkers of hypoxia have been extensively validated in multiple cohorts of patients undergoing EBRT, predicting prognosis and benefit of hypoxia modification [6]. Transcriptomic biomarkers of intrinsic radiosensitivity have also been validated in the photon EBRT setting in patient cohorts [7] but fully validated gene expression signatures for MRT are limited.

The exposure of normal tissue, including kidney and liver tissue, during MRT cancer treatment is problematic. One example is the accumulation of somatostatin-targeting treatments in the kidney due to the expression of somatostatin receptors and proximal tubule reabsorption [8], as well as salivary gland toxicity in patients treated with radiolabelled PSMA ligands [9]. To inform on clinical MRT procedures pre-treatment, dosimetry and robust biomarkers of radiosensitivity are required. Here, transcriptomic and a few illustrative imaging studies during MRT are reviewed, which will be helpful in identifying radiation responsive genes that may be predictive of MRT response and normal tissue toxicity.

### 1.2. Radionuclides Used in MRT

Radionuclides delivered by molecular radiotherapy are mainly β and α emitters. β-emissions have a longer range in tissue (up to 10 mm) and a low linear energy transfer (LET) which means that the density of ionisations, and the subsequent damage to molecules (particularly DNA) they induce, is relatively sparse. α-emissions have short ranges (up to 100 µm or 1–4 cell diameters in tissue) but high LET imparting high levels of damage, including many more lethal DNA double-strand breaks within a short range. Most licensed MRT radiopharmaceuticals use β-emitters, but there are now many trials of α-labelled radiopharmaceuticals to take advantage of the greater lethality of high LET radiation.

Suitable β- and α-emitting radionuclides include [^177^Lu], [^131^I], [^90^Y], [^161^Tb], [^225^Ac], [^211^At], [^212^Bi], [^213^Bi], [^212^Pb], [^223^Ra], [^149^Tb], and [^227^Th]. Some radionuclides decay through a mixture of α and β emissions, which can be effective in delivering local high LET radiation and longer-range crossfire to cancer cells distant from the emitting atom to improve dose-distribution across a tumour.

Neuroendocrine tumours (NET), a highly heterogeneous and rare cancer type (<1% of all malignancies), overexpress somatostatin receptors (SSTR). DOTA-tyrosine^3^-octreotate (DOTATATE) is a ligand of SSTR and binding results in its cellular internalisation. Prostate specific membrane antigen (PSMA), a type II transmembrane glycoprotein, is overexpressed by the majority (>90%) of prostate cancers, especially advanced stage disease [10]. Prostate cancer can be targeted with the PSMA ligand PSMA-617 and with the antibody J591. NETs can be treated with versions of DOTATATE labelled with beta-emitters, including ^177^Lu and ^90^Y, an example of a peptide receptor radionuclide therapy (PRRT). PSMA-617, labelled with [^177^Lu], a medium energy β-emitter (0.49 MeV) with a range in tissue of up to about 2 mm, is also used in the treatment of mCRPC. Accompanying γ-emissions allow for image acquisition and dosimetry with SPECT/CT [11].

[^225^Ac] decays with a cascade of 6 short-lived daughter radionuclides comprising 4 α-particles, with a range in tissue of up to 85 µm (about 4 cell diameters), and two β’s with tissue range of up to about 8 mm. Toxicity issues with both [^177^Lu]-PSMA-617 and particularly with [^225^Ac]-PMSA-617, including kidney and xerostomia due to salivary gland uptake [9], have led to an interest in the use of the radiolabelled versions of the antibody J591 for treating prostate cancer, which is less tissue penetrating than the smaller ligands PSMA-617, potentially reducing toxicity associated with normal tissue uptake [12]. Combination treatment with both [^225^Ac] and [^177^Lu]-PSMA-617 demonstrate low toxicity but these studies only used a single cycle of each radiopharmaceutical [13,14].

## 2. Imaging of Patients Undergoing MRT

The imaging modalities single-photon emission computer tomography (SPECT) and positron emission tomography (PET) which, respectively, detect gamma or positron emitting radionuclides conjugated to antibodies or peptides with high affinity for target receptors, can be used to determine the level of receptor expression. This will determine patient suitability for treatment with MRT and enables dosimetry to tailor activity levels on a patient-by-patient basis [15,16].

Imaging outputs generally include measures of mean and maximum standardized uptake (SUV_mean_ and SUV_max_) by a lesion. Other parameters include estimates of dose heterogeneity across lesions. The glucose analogue, [^18^F]-Fluoro-2-deoxy-D-glucose (FDG), used in >80% of PET investigations, exploits high glucose utilisation by cancer cells. Well-differentiated thyroid tumours will concentrate iodine, which is the basis of [^131^I]-therapy, a property lost as the cancer dedifferentiates. Dedifferentiation is accompanied by a decreased glycolysis detectable by FDG-PET [17]. However, an altered glucose metabolism accompanies many cancer processes, including hypoxia [18], making FDG too general a tracer to be useful as an indicator of tumour differentiation.

The response of NETs to PRRT is highly variable, with an overall successful outcome in about 60% of patients [19]. DOTATATE can be labelled with positron-emitting radioisotopes, including ^64^Cu and ^68^Ga, for PET imaging. A recent pilot study suggests that the heterogeneity of distribution of [^68^Ga]-DOTATATE, by NETs in patients who are then treated with [^177^Lu]-Lu-DOTATATE, corresponds with patient outcome (progression free survival(PFS)) [20]. Pre-treatment scans with [^68^Ga]-Ga-DOTA-TOC of patients with NETs have enabled the prediction of the absorbed doses of the subsequent [^177^Lu]-Lu-PRRT to be determined [21]. However, the associations between SUV and patient outcome are not always consistent between studies due to a current lack of standardisation [22].

The PET radiolabelled ligand PSMA-617 [^68^Ga]-PSMA-617 is approved for prostate cancer imaging. However, about 30% of patients with prostate cancer overexpressing PSMA do not demonstrate a biochemical response (reduction in serum prostate specific antigen (PSA) level) to [^177^Lu]-PSMA-617 treatment [2].

## 3. Gene Expression and Mutation

### 3.1. Clinical Transcriptomic Studies

The first application of radionuclide therapy in cancer treatment was the use of [^131^I]-Iodide to treat iodine-avid thyroid cancers [23] during the 1940s. To better understand the therapeutic mechanism key genes that respond to [^131^I]-Iodide treatment were determined by comparing thyroid cancers from patients undergoing [^131^I]-Iodide or by other therapies using publicly available RNA-seq data [24]. Genes differentially regulated underwent pathway enrichment analysis from which *CDH5*, *KDR*, *CD34*, *FLT4*, *EMCN*, *FLT1*, *ROBO4*, *PTPRB*, and *CD93* were the most important hub genes and are predominantly associated with vascular function. The expression of the hub genes in patients treated with ^131^I-iodide was verified using PCR. Shuwen et al. used *GAPDH* as an ‘internal control’ which may be unreliable as it is sensitive to hypoxia, found to varying degrees in most solid tumours [25,26].

Table 1 shows the main genes consistently modulated in tissue samples from patients treated with radionuclide therapy. These tissues include blood samples from which DNA transcripts or circulating tumour cells (CTCs) can be analysed. Circulating tumour cells (CTCs) are a source of ‘real-time’ genetic information from tumours and can be used to follow the evolution of the cancer including during treatment. Transcriptomic profiling on a microarray consisting of 64 genes highly expressed in prostate cancer with low expression in leukocytes (to avoid input from contaminating blood cells) was carried out on lysed CTCs from 40 patients at baseline (prior to treatment) and 29 normal controls to determine cancer gene specificity [27]. Treatment modalities were monotherapy or combinations of chemotherapy, anti-androgen, radioligand, immunotherapy and PARPi. A hierarchical clustering of patients by gene expression independently of treatment demonstrated two groups. A high expression of androgen-receptor signalling associated genes (*ARv7*, *DLX1*, *HOXB13*, and *KLK3*) DNA damage repair genes (*BRCA1*, *BRCA2*, *FANCA* and *TOP2A*) and the oncogenes *ERG* and *GLRH2* was associated with reduced overall survival.

PSMA expression on the cell membrane of both primary and metastatic lesions is essential for response to PSMA-targeted treatments. Epigenetic modifications were found to modulate PSMA expression in cancer lesions [29]. Sayer et al. [29] determined the expression of PSMA across 339 lesions from 52 patients with mCRPC. PSMA protein expression determined by immunohistochemistry was highly correlated with gene expression level (FOLH1) across the 339 lesions. mCRPC can be divided into four molecular subtypes dependent on androgen receptor and neuroendocrine (NE) marker expression [36]. PSMA expression was highly variable across the four molecular subtypes. Intra-tumour heterogeneity in PSMA expression between different lesions from the same patient was evident in almost 50% of patients. PSMA expression was also dependent on the anatomical site of the lesion, with lower expression in bone metastasis compared with liver. Intra-tumour heterogeneity was also evident, with distinct regions of low and high PSMA expression within individual lesions. Sayer et al. [29] found that differences in PSMA expression were associated with increased CpG methylation and decreased histone lysine 27 acetylation, and that targeting epigenetics increased the expression of PSMA in an in vivo model of prostate cancer. This paper demonstrates the importance of looking at methylation/acetylation as additional components of MRT response signatures.

[^131^I]-mIBG is an analogue of norepinephrine and accumulates in neuroblastomas and phaeochromocytomas. To identify genes that may predict absorbed dose, blood samples pre-, 72 h and 96 h post-treatment were obtained from 40 patients treated with mIBG [30]. Using the PCR expression of ten genes representative of tumour suppressor protein p53 apoptosis pathways, including *FDXR*, *BBC3*, *BCL2*, *BCLXL*, *BAX* and *BIM*, cell cycle arrest domain *CDKN1A* and *GADD45A* and DNA repair *XPC* and *DDB2* were determined. The transcripts *CDKN1A*, *FDXR*, *GADD45A*, *STAT5B*, *BAX*, *XPC*, *MDM2* and *DDB2* were significantly upregulated at 72 h and 96 h post-treatment. BCLXL was significantly downregulated. The *BCL2* transcript did not show modulated expression. The absorbed dose was determined by dose-retention measurements at the tumour site carried out throughout the 96 h. A gene expression biodosimetry model was developed to predict absorbed dose based on the modulation of gene transcripts within whole blood. Most of the variance in the modulation of gene expression over the 96 h was due to three genes, CDKN1A, BAX and DDB2.

More recently, the group has [31] extended the post-treatment time to 15 d and to 59 patients. Changes in gene expression at 72 h largely confirmed their earlier study, except BCL2 expression, which was significantly decreased at 72 h but unchanged in their previous study. This may reflect increased powering of the study with a larger patient cohort. The transcripts *CDKN1A*, *FDXR*, *DDB2*, and *BBC3* demonstrated very significant (*p* < 0.000001) up-regulation at 72 h after [^131^I]-mIBG exposure. The transcripts *FDXR* and *DDB2* increased in expression at both 72 h and 15 d whilst *BCL2* and *SESN1* maintained decreased expression 15 days after [^131^I]-mIBG treatment. The transcripts *XPC*, *STAT5B*, *PRKDC*, *MDM2*, *POLH*, *IGF1R*, and *SGK1* displayed significant up-regulation at 72 h and significant down-regulation at day 15, suggesting that transcript levels for DNA repair, apoptosis, and ionizing radiation-induced cellular stress are still changing 15 days after [^131^I]-mIBG treatment. At 15 d the absorbed dose could not be determined using a gene-expression biodosimetry model but could identify if patients had received ^131^I-mIBG. These two studies [30,31] used *GAPDH* as the endogenous control, which may demonstrate some expression variation between tumours.

Kidd et al. [32] developed the NETest which comprises the transcriptomic expression of 51 genes involved in neoplastic behaviour associated with the NET proliferation, signalling and secretion present in the blood of patients with NETs. A comparison was carried out between circulating and tumour tissue or normal mucosa from 9 patients and 7 normal controls, respectively. NET scores were highly correlated between tissue and blood samples. The NETest, using circulating neuroendocrine tumour transcripts, differentiated stable disease from progressive disease.

Bodia et al. [33] compared NETest with a PRRT predictive quotient (PPQ) test in an interim study of a cohort of patients with gastroenteropancreatic and bronchopulmonary NETs who had completed PRRT treatment. The PPQ was based on tumour grade and the levels of mRNA in the growth factor-related genes *ARAF*, *BRAF*, *KRAS and RAF-1* and genes involved in metabolism *ATP6V1H*, *OAZ2*, *PANK2* and *PLD3* in blood samples, normalized to expression of ALG9 and applied categorically. The PPQ was validated in three cohorts of patients with NETs treated with [^90^Y] or [^177^Lu]-PRRT [34]. Forty patients were PPQ+, of whom 39 responded to PRRT. Of the 27 PPQ- patients, 25 progressed. The overall predictive accuracy was 96% (*p* < 0.0001) for PPQ status and 90% for the NETest.

Bodia et al. [33] also examined mutations in tumour tissue from 30 patients. The most common alterations were in the mTOR/PTEN/TSC pathway (n = 9) and MEN-1 (n = 8), but no relationship between mutations and outcome was found. Genetic alterations were explored in a small study by Satapathy et al. [35] comparing the outcomes of 15 patients with mCRPC with different genetic alterations treated with either [^177^Lu]-Lu-PSMA-617 (n = 12) or [^225^Ac]Ac-PSMA-617 (n = 3). In 10/15 (67%) patients, 21 genetic alterations including 13 DDR-associated alterations involving the genes *ATM*, *BRCA2*, *TP53*, *PTEN*, *FANCD2*, *FANCM* and *NBN* were found. Overall, the tumours from 5/15 (33%) patients possessed pathogenic variants (*BRCA2*, *ATM*, *TP53*, and *PTEN*). No significant difference was noted for the biochemical response, radiological response, PFS, and overall survival between the patients with and without genetic alterations. The small patient numbers in which mutation status was determined hinder any definitive conclusions from these two studies [33,35].

Approaches to improving the response of patients with mCRPC to PSMA-targeted MRT include combination treatment with immune checkpoint inhibitors [37]. A transcriptomic study using RNA obtained from diagnostic and excision blocks from 23 patients treated for prostate cancer with [^177^Lu]-PSMA-617 investigated microtumour environment gene-expression signatures to predict outcomes [28]. They found that the PD-L2 signature was significantly associated with overall survival, suggesting crosstalk between the immune microenvironment and radiation efficiency. However, prostate tumours are known to be an immunologically ‘cold’ cancer type in which immunotherapy benefit may be very limited [38].

### 3.2. Preclinical Transcriptomic Studies

Table 2 shows the genes modulated by exposure to clinically relevant radionuclides in normal and tumour tissue in preclinical studies.

#### 3.2.1. Normal Tissue Absorbed Dose and Gene Expression

Astatine-211, [^211^At], which has a t_1/2_ = 7.2 h and decays by α-emission and electron capture, has been proposed as an alternative to ^131^I for the treatment of medullary thyroid carcinoma due to its accumulation in thyroid tissue [48]. However, ^211^At also accumulates in other normal tissues including kidney, liver, lung and spleen [44]. Langen et al. [44] determined organ absorbed dose and RNA expression in BALB/c nude mice 24 h after being administered with solutions of 5 doses of ^211^At (0.064–42 kBq) in phosphate-buffered saline (PBS). A control group received a ‘mock injection’, which would usually be a normal injection of vehicle (PBS in this case), but this is not stated in the paper. The absorbed dose, linearly related to the administered dose, was 130, 160, 300 and 1000 mGy in the liver, kidney, spleen, lungs, respectively, after the administration of 42kBq. Interestingly, the number of differentially expressed, particularly upregulated genes, dramatically decreased as the ^211^At dose increased at a threshold between 0.64kBq and 1.8kBq, indicative of a dose-dependent dichotomy in transcriptional control. Only one gene, ANGPTL4 (angiopoietin like protein 4) was consistently altered in expression across most organs and doses. Out of a panel of 56 genes previously proposed as radiation-sensitive [49,50], 37 showed no change in any tissues and only 19 showed a change in 1 tissue or more. Some differential expression of DNA integrity genes was identified but genes for DNA damage and repair were only differentially expressed in spleen. The upregulation of RGS16 (regulator of G-protein signalling 16) increased with an increased absorbed dose from 2.0 mGy to higher absorbed doses and may be a suitable biomarker of liver response to radiation.

Schuler examined mRNA expression in kidney tissue from mice for 24 h [45] and 12 months [46] after the administration of 5 doses (1.3, 3.6, 14, 45 and 140 MBq) of [^177^Lu]-octreotide. Determined using Medical Internal Radiation Dose (MIRD), these equated to absorbed doses of 0.13 to 13Gy. Using microarrays, Schuler identified 32 and 39 transcripts in the kidney cortex and medulla, respectively, consistently regulated in all 5 doses with 22 common to both regions. *ANGPTL4* (angiopoietin-like 4) was consistently upregulated except at the highest dose. A strong association to metabolism was found among the affected processes in both tissues associated with cellular and developmental processes, which was prominent in the kidney medulla, while transport and immune response were prominent in kidney cortex. To verify the microarray results, nine significant upregulated genes were detected in both renal tissues at all absorbed dose levels (*ANGPTI4*, *CYP24A1*, *DNASE1*, *GLDC*, *HMGCS2*, *IGFBP4*, *S100A8*, *SLC25A20*, and *SLC25A25*) were determined by PCR. *CYP24A1* has been previously shown to be increased in expression in the kidney tissue of mice exposed to ^131^I [47] and *ANGPTI4* in ^211^At exposed tissues [44], suggesting some crossover between tissue response to different types of radiation. Twelve months after the treatment, the glomerular filtration rates, measured by the excretion of [^99m^Tc]-DTPA, were diminished in a dose-dependent manner compared with pretreatment [46]. Gene expression of *CDKN1A*, *C3*, *DBP*, *LCN2*, and *PER2* genes displayed an absorbed dose-dependent increase in expression, suggesting that they can be used as biomarkers of exposure.

MicroRNAs (miRNA) are small 18–22nucleotide RNA molecules that bind to the 3’ UTR (untranslated regions of mRNA) interfering with their translation. They can bind to many different mRNAs, influencing the translation of several gene transcripts. MicroRNAs have a strong influence on the response to DNA damage [51]. Schuler et al. [45] determined the expression of miRNAs in the same mice as for their mRNA study and found that in total, across all four doses of [^177^Lu]-octreotide, 57 miRNAs were differentially regulated (mainly upregulated) in treated mice. Only 4 miRNAs were commonly consistently differentially regulated (all upregulated at the four higher doses): miR-194, miR-107, miR-3090 and miR-3077. This is consistent with previous studies, which have demonstrated that the response to ionizing radiation is highly dependent on dose, timing and cell type, calling into question the utility of miRNAs as general biomarkers with respect to normal tissue radiation exposure [52].

#### 3.2.2. Tumour Tissue

To identify potential markers of response to [^211^At]-labelled complexes, Ohshima et al. [40] compared RNAseq data from phaeochromocytoma cells, a neuroblastoma cancer type, incubated with meta-[^211^At]-astato-benzylguanidine (^211^At-MABG) and γ-irradiated cells using a ^60^Co source. Gene expression data were obtained from cells 0, 3, 6 and 12 h post-treatment. To equalise the biological effect between the two radiation types, iso-survival doses (10% (LD90) and 80% (LD20)) were derived from dose–survival curves. Genes associated with p53 signalling pathways were increased in expression, including *MDM2*, *CDKN1A*, *GADD45A*, *GADD45C* and *RB1* genes, which were increased in both ^211^At-MABG and γ-radiation. Genes associated with decreased survival (genes that increased in expression between cells treated with LD20 and LD90 doses) were upregulated at all times and included activated p53 target genes, *GDF15*, *FAM212B*, *CDKN1A*, *ENC1* and *TP53INP1*. The homologous recombination gene (HR), *FAM175A*, was upregulated at 3 h post-treatment. Genes specifically responsive to [^211^At]-MABG treatment included the genes associated with the development of metastasis *MIEN1* and *OTUB1*, the angiogenesis growth factor *VEGFA*, and *VDAC1*, which regulates nutrient transport across the outer mitochondrial membrane and is associated with apoptosis. This study confirms the role of p53 in the radiation response to MRT, and the commonality and differences in genes expressed by cancer responding to different forms of ionising radiation.

Spetz et al. [42] determined gene expression changes in somatostatin receptor (SSTR)-expressing (GOT) tumours in mice treated with a pre-priming schedule of a 5 MBq then 1 d later a 10 MBq dose of [^177^Lu]-Octreotide. Tumours were harvested at 1, 3, 7 and 41 d post second injection. Transcriptomic analysis identified 187 transcripts differentially regulated. At 1 d, 3 d, 7 d and 41 d, 33 (66%), 41(60%), 48 (59%), 27 (87%) were uniquely differentially regulated. Thirty-eight genes were differentially regulated at at least two time points. Interestingly, the genes associated with maintenance of DNA integrity were only differentially regulated at 3 d, whilst the genes associated with stress response were differentially regulated at 3 and 7 d. The priming schedule induced cell death by both intrinsic (including *BAX*, *GADD45A*, and *PBK*) and extrinsic (including *TNFRSF10B* and *NGFRAP1*) pathways. At a point late in time, the inhibition of proliferation is indicated by the gene downregulation of *CXCR7* and *LGALS1* genes. A comparison with an earlier study, where 15 MBq was delivered as a monotherapy [41], demonstrated the distinct expression of a p53-cell cycle arrest and extrinsically induced apoptosis, and induction and PI3K/PKT regulation were specific to the pre-priming approach. The findings from this study demonstrate (1) unique time dependent gene expressions with some commonality and (2) distinctions in the delivery schedule of radionuclides can have a differential effect on gene expression, which needs to be considered when developing transcriptome based MRT response biomarkers.

The upregulation of the PI3K/PKT pathway underlies some of the resistance mechanisms associated the treatment of HER-2 overexpressing breast cancer and colorectal cancers with the antibody Trastuzumab [53]. However, HER-2 can still be an effective target for the MRT targeting. Yong et al. [54,55,56] conjugated Trastuzumab with the α-emitter [^212^Pb]. Mice bearing xenografts, derived from colorectal cancer cells, were injected with the [^212^Pb]-Trastuzumab antibody or a non-specific antibody, [^212^Pb]-HuIgG, with paclitaxel, which has been shown to potentiate the efficacy of [^212^Pb]-Trastuzumab [54] or with Gemcitabine [55]. Gene expression analysis was carried out using PCR with an 84 gene DNA damage array on xenografts harvested 24 h after treatment. The genes upregulated by [^212^Pb]-Trastuzumab, *ABL*, *MKK6*, *SESN1* and *BTG2*, were also upregulated and DDIT3 was downregulated by [^212^Pb]HuIgG [56]. Many genes modulated in expression when [^212^Pb]-Trastuzumab is combined with paclitaxel are modulated by paclitaxel alone or [^212^Pb]-HuIgG/Pac genes but to a lesser extent [54]. Genes associated with apoptosis, except BRCA1 and p73, were modulated by Gem alone as well as [^212^Pb]Trastuzumab/Gem, suggesting Gemcitabine has the greatest impact on gene expression. Antimetabolites have previously been shown to induce apoptosis [57]. [^212^Pb] exposure (conjugated with HuIgG or Trast) combined with Gem downregulated BRCA1 and p73, suggesting surprisingly that the targeting of the [^212^Pb] binding, which would affect tumour dose, is not important. The genes associated with cell cycle regulation that were most significantly increased by [^212^Pb]-Trast/Gem were *CHK1*, *CHK2*, *FANCG*. The remaining genes were equally modulated by Gem alone. Genes associated with DNA damage and repair modulated by [^212^Pb]-Trast/Gem or [^212^Pb]-HuIgG/Gem but not Gem alone were *ERCC1* and p37, which were upregulated, and *BRCA1*, *FANCG*, *FEN1*, *RAD18*, *RAD51a*, *UNG* and *XRCC2*, which were downregulated.

The findings from these three studies, summarised in Table 3, demonstrate commonality in the modulation of expression of some genes by tumours treated with DNA-damaging chemotherapy drugs and with α-emitters. Combination treatments enhance the expression of radiation/chemotherapy responsive genes. Some genes do appear to be specifically modulated by [^212^Pb] exposure.

Some gastric cancers overexpress the cadherin and d9-E-cad, and can be efficiently targeted with the antibody d9Mab. Seidl et al. [43] exposed a d9-E-Cad overexpressing gastric cancer cell line to d9Mab conjugated to Bismuth-213 (^213^Bi), an α-emitter with a t_1/2_ of 46 min. A microarray and a confirmatory PCR was carried out on control and cells exposed to ^213^Bi-d9Mab (1.5 MBq/mL) for 6, 24 and 48 h. Cell death was observed between 72 and 96 h of treatment. The microarray demonstrated that between 666 and 1278 genes were downregulated and between 682 and 1125 genes upregulated by exposure to [^213^Bi] at the three time points. However, only 8 and 12 genes were upregulated or downregulated, respectively, at all three time points, which indicated time-dependence on gene expression. The consistently upregulated genes *COL4A2*, *NEDD9*, *C3*, *NMES1* and *RASA4* were involved in cell adhesion, cell cycle control, complement-mediated immunity, carbon metabolism and signal transduction, respectively. Genes that were consistently downregulated comprised cell cycle control/cytokinesis (*CDC20*, *GAS2L3*, *ASPM*) chromosome segregation (*KIF20A*, *CENPE*), chromatin packaging (*HIST4H4*), proteolysis (*WWP2*) and transcription regulation (*RFX3*, *PHF17*). Knockdown for some of these genes has previously been shown to be catastrophic for the cell, consistent with cell death induced by exposure to [^213^Bi], e.g., knockdown of *CENPE* [58] or *ASPM* [59] results in mitotic spindle disturbances.

In view of the limitations of PSMA-targeted MRT, which are kidney and salivary gland toxicity and a lack of response in about 30% of patients with mCRPC, other targets are being explored to treat this cancer, including androgen-receptor regulated proteins [39]. Human kallikrein-related peptidase 2 (hK2), an AR-regulated prostate enzyme, is abundant in some prostate cancers even where PSMA expression is diminished [39]. Xenografts derived from LNCaP-AR prostate cancer cells in mice treated with [^225^Ac]-hu11B6, a hK2-targeting platform, demonstrated complete or partial regression, but was followed by a relapse in some mice [39]. An RNA analysis of excised xenografts identified 1644 genes differentially expressed (512 up-, 1132 downregulated) in the relapsing tumours. The upregulated genes included *MMP7*, a matrix metalloprotease associated with ab increased risk of metastasis [60], *ETV1*, which is linked with loss of the tumour suppressor gene PTEN [61], NTS associated with activation of the Akt pathway [62], *SCHLAP1*, and androgen repressed genes, including *PMP22*, *CAMK2N1*, *UGT2-B17UDP*. The downregulated genes included the AR-regulated genes *KLK3*, *FOLH1*, *PMEPA1*, *SPOCK1*.

## 4. Discussion

The nuclear medicine imaging of patients using the version of MRT agents labelled with γ or Positron-emitters provides essential information on receptor status and therapeutic binding, but it does not necessarily predict response. Transcriptomics can provide an insight into the biology of individual cancers, which may predict their response and inform dose tailoring as for EBRT using genome assisted radiotherapy (GARD) [7,63]. The rationale for this review was to bring together the findings from preclinical and clinical studies, which have determined gene expression in tissue from patients/biological models receiving radionuclides, to inform on the development of response biomarkers. In view of the inclusion of radiological features with genomics in generating robust biomarkers, known as radio-genomics, the review also included a small section on imaging in MRT.

Several transcriptomic studies of tissues from patients with NETs treated with MRT have identified potentially useful gene panels that predict outcomes, though only in small patient cohorts and with relatively short census times [32,33]. A few transcriptomic studies described have determined the predictive capability of gene expression pre-treatment, rather than most studies, which have looked at changes in expression during treatment. These will enable the assessment of suitability for continued treatment but, in identifying pathways that are important in response, they may provide seed genes for the generation of predictive transcriptomic biomarkers of MRT.

Taken together, these studies have provided insight into the complexities of response to radionuclides demonstrating that gene expression is dependent on factors including: time after treatment—unsurprising, as processes that are activated in response to radiation damage occur in a time-ordered sequence; dose scheduling of radionuclide treatment; type of radiation—some commonalities in genes expressed in response to different radiation types were identified, and also with chemical DNA damaging agents including genes involved in response to DNA damaging agents such as BRCA1; the dependency of gene expression on epigenetics, supporting the inclusion of epigenetic markers in gene signatures.

Several genes were commonly identified in different studies including *CDKN1A*, *GADD45A*, *GADD45C*, *ANGPT14*, and activated p53 responsive genes such as MDM2. Many of the genes modulated by exposure to radionuclides would not necessarily be predicted as radiation regulatable, demonstrating the importance of experimental studies to identify radiation-responsive genes. None of the genes in the GARD radiation sensitivity index (RSI) [7] were modulated by exposure to any of the MRT radionuclides in the cited studies. However, as outlined below, the process for deriving a prognostic/predictive gene expression from an initial candidate list involves multiple selection steps.

Biomarkers based on gene signatures can be defined as a set of genes for which the collective changed expression has been validated to demonstrate diagnosis, prognosis or to predict therapeutic response. Gene expression signatures can be derived by a sequence of steps involving the identification of seed genes, co-expression network analysis, gene selection on prognostication using a clinical cohort (test) and finally validation in additional patient cohorts [63]. Seed genes can be identified from studies demonstrating expression modulation by therapeutic exposure [64]. Alternatively, they can be identified through the exposure of different cancer cell lines to the therapy of interest and the identification of genes that correlate with therapeutic efficacy [65]. The sample number was low for some studies cited, for example, Kidd et al. [32] demonstrated a strong correlation between gene expression scores determined from tumour tissue and circulating RNA in 9 patients. Handke et al. [28] determine RNA expression in only 23 patients. As more cancer patients are prescribed MRT, there will be opportunities for gene transcriptomic studies to be carried out on larger patient populations particularly of prostate cancer patients. This will enable the confirmation of findings from smaller studies. There are also initiatives to create cross-nation interest groups in MRT, such as the recent collaboration between UK and Dutch universities with the publication of a white paper [66], which will enhance tissue availability between participating groups.

The studies cited in this review illustrate the derivation of genomic information from liquid biopsies and tumour tissues. Sumanasuriya et al. [67] have demonstrated that ctDNA-generated genomic biomarkers derived from patients with mCRPC can be prognostic. Poor levels of concordance in genomic alterations between tumour tissue and liquid biopsy-derived tissue [68], low sensitivity of liquid biopsy-derived material for identifying genomic alterations [69] and poor stability of RNA in serum may be limiting [70]. Despite these issues, tests (NETest [32] and PPQ [33]) generated from genomic information from liquid biopsy material have been shown to be predictive of response to PRRT.

Gene expression data can be generated on different platforms including PCR and microarray [71], and their selection should consider future clinical translation. Endogenous (or housekeeping) gene selection underpins the accuracy of PCR, and their invariant expression should be determined on study samples [26]. The GAPDH gene, which is hypoxia-modulated [25] and may vary in expression between tumours [72], was used in several studies [24,30,31].

In summary, studies have shown that cells/tissues treated with medically relevant radionuclides have common and unique gene expression changes dependent on many factors, including sampling time-post treatment, radiation-type and fractionation, tissue/tumour type. Nucleic acids derived from tumour tissue/normal tissue and liquid biopsies are potential sources of information on gene expression/DNA alterations to derive prognostic and predictive biomarkers.

## Figures and Tables

**Figure 1 genes-15-00688-f001:**
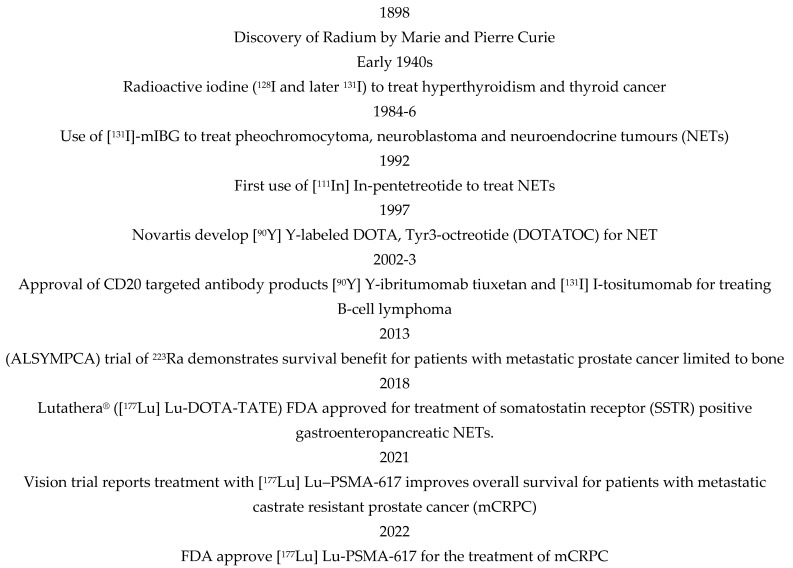
Notable dates in the application of molecular radiotherapy (MRT).

**Table 1 genes-15-00688-t001:** Transcripts consistently modulated in their expression in tumour tissue, and liquid biopsies from patients treated with radiation.

Tissue	Treatment	Main Findings/Genes Modulated	Study
Prostate tumour n = 23 (transcriptome)	[^177^Lu]-PSMA-617	PD-L1 signature genes	[28]
Blood CTCs n = 40 (MRT = 2)	All treatment regimens for prostate cancer	Genes associated with poor outcome: *ARV7*, *DLX1*, *HOXB13*, *KLK3* (Androgen receptor signalling) *BRCA1*, *BRCA2*, *FANCA*, *TOP2A* (damaged DNA response) and *ERG* and *GLRH2* (oncogenes)	[27]
mCRPC primary and metastasis n = 52	[^177^Lu]-PSMA-617	Low PSMA expressing tumours include: ↑*DHRS9*, *JAK3*, *MATK*, *PTGES*, *ADH1C*, *MAPK15*, *CDK6*, *MUC1*, *EMA*, *MSLN*, *CEACAM5* PSMA expression linked to ↑CpG methylation and ↓histone Lys acetylation	[29]
Blood NET n = 40	[^131^I]-mIBG	Post treatment at 72 h and 96 h: ↑*CDKN1A*, *FDXR*, *GADD45A*, *STAT5B*, *BAX*, *XPC*, *MDM2*, *DDB2* Predictors of absorbed dose: *CDKN1A*, *BAX*, *DDB2*	[30]
Blood NET n = 59	[^131^I]-mIBG	Post treatment 72 h and 15 d: ↑*FDXR* and *DDB2* at 72 h and 15 d At 15 d absorbed dose could no longer be determined but exposure to [^131^I]-mIBG still evident in gene expression.	[31]
Blood NET n = 130 test n = 159 tumour n = 22	[^177^Lu]-DOTATATE	NETest 51 genes involved in neoplastic behaviour in NETs. Differentiated patients with progressive and stable disease	[32]
Blood NET n = 158	[^90^Y]-or [^177^Lu]-DOTATATE	PRRT-predictive quotient (PPQ): tumour grade+ mRNA levels: Growth factor *ARAF*, *BRAF*, *KRAS*, *RAF-1* and metabolism *ATP6V1H*, *OAZ2*, *PANK2*, *PLD3* Outcome prediction accuracy: PPQ: 96% NETest 90%	[33,34]
Blood n = 15	[^177^Lu]- or [^225^Ac]- PSMA-617	21 genetic alterations involving *ATM*, *BRCA2*, *TP53*, *PTEN*, *FANCD2*, *FANCM*, *NBN* no relationship with outcome	[35]

mCRPC metastatic castrate-resistant prostatic cancer; NET neuroendocrine tumour; n = number of patients used in analysis; ↑,↓ up- or down-regulated

**Table 2 genes-15-00688-t002:** Genes consistently modulated in expression in biological models by exposure to clinically relevant radionuclides (preclinical studies).

Tissue	Radiation Type/Compound	Gene Expression	Study
Prostate xenograft	^225^Ac (11kBq) Kallikrein-related peptidase 2 targeted molecule	Tumours that relapsed after ^225^Ac: Up: *MMP7*, *ETV1*, *NTS*, *TMEFF2*, *SCHLAP1*, *PMP22*, *CAMK2N1*, *UGT2-B17UDP* Down: *KLK3*, *FOLH1*, *PMEPA1*, *SPOCK1*	[39]
Phaeochromocytoma	^211^At (0.85kBq) compared with γ (^60^Co) m-astato-benzylguanidine	Consistently expressed genes all times: Both α + γ: P53 associated: MDM2, *CDKN1A*, *GADD45A*, *GADD45C*, *RB1* Genes related to decreased survival: *GDF15*, *FAM212B*, *CDKN1A*, *ENC1*, Genes specific to survival after ^211^At: *MIEN1*, *OTUB1*, *VEGFA*, *VDAC1*, *P53INP1* (all are activated p53 targets)	[40]
Intestinal neuroendocrine tumour (GOT) xenograft	^177^Lu 15 MBq DOTATATE	All time points: *LY6H*, *RNU1A3*, *RNU1-5* Early (1 d) ↑*CDKN1A*, *IL6*, *TGFβ1*, *HIF1*↓*BCAT1*, *PAM* 3 d:↑*APOE*, *DAX* (apoptosis intrinsic p/w) 41 d: ↑*ADORA2A*, *BNIP3*, *BNIP3L*, *HSPB1*, *NEDD9*, *TNF* Stromal: *ACTA1*, *SERPIN3g*, *ATP2A1*, *CXCL9*, *TNN12*, *MYL1*, *MYLPF*	[41]
GOT xenograft	^177^Lu 5 then 10 MBq DOTATATE	In addition to 15 MBq: ↑*TNFRSF10B*, *NGFRAP1* (apoptosis extrinsic p/w) ↓*CXCR7*, *LGALS1* (proliferation) corresponds with slower regrowth with pre-priming	[42]
Gastric cancer cells	^213^Bi (1.5 MBq) Cadherin antibody d9-E-Cad	↑*COL4A2*, *NEDD9*, *NMES1*, *RASA4* ↓*CDC20*, *GAS2L3*, *ASPM*, *KIF20A*, *CENPE*, *HIST4H4*, *WWP2*, *RFX3*, *PHF17*	[43]
Mouse liver, kidney, lung, spleen	^211^At (0.06–0.42 kBq)	↑*ANGPT14* all organs and doses ↑*RGS16* with absorbed dose by liver Transcriptional response highest at low absorbed dose; greatly decreased at higher doses. Above threshold ↑Amy2 in spleen	[44]
Mouse kidney	^177^Lu (1.3–140 MBq) Octreotide	24 h post treatment ↑*ANGPT14* all except highest dose and *CYP24A1*, *DNASE1*, *GLDC*, *HMGCS2*, *IGFBP4*, *S100A8*, *SLC25A20*, *SLC25A25* altered at some doses. Across all doses microRNAs consistently ↑miR-194, miR-107, MiR-3090, miR-3077 12months post treatment–kidney function diminished and ↑*CDKN1A*, *C3*, *DBP*, *LCN2* and *PER2*	[45,46,47]

↑, ↓ up- down-regulated genes.

**Table 3 genes-15-00688-t003:** Genes up- or downregulated in xenografts harvested from mice 24 h after treatment with [^212^Pb]Trastuzumab alone or in combination with chemotherapy agents paclitaxel or Gemcitabine. Data taken from Yong et al. [54,55,56].

Treatment	Category	Gene Upregulated	Gene Downregulated
[^212^Pb]-Trastuzumab	Apoptosis	*ABL*, *GADD45A*, *GADD45C*, *PCBP4*, *p73*	
[^212^Pb]-Trastuzumab/Paclitaxel		*GADD45A*, *GADD45C*, *PCBP4*, *P73*, *GML*, *IP6K3*, *PPP1R15A*, *CIDEA*	BRCA1, RAD21
[^212^Pb]-Trastuzumab/Gemcitabine		*GADD45A*, *GADD45C*, *PCBP4*, *P73*, *IP6K3*	*BRCA1*, *RAD21*
[^212^Pb]-Trastuzumab	Cell cycle	*ATM*, *GADD45A*, *PCBP4*, *MKK6*, *SESN1*, *ZAK*	*DDIT3*, *GTSE1*
[^212^Pb]-Trastuzumab/Paclitaxel		*GADD45A*, *GML*, *PCBP4*, *PPP1R15A*, *RAD9A*, *SESN1*	*BRCA1 and RAD21 CHK1*, *CHK2 GTSE1*, *NBN*
[^212^Pb]-Trastuzumab/Gemcitabine		*GADD45A*, *MAP2K6*, *PCBP4*, *SESN1*	*CHK1*, *CHK2*, *FANCG*, *GTSE1*, *NBN*, *BRCA1*
[^212^Pb]-Trastuzumab	DNA damage and repair	*ATM*, *BTG2*	
[^212^Pb]-Trastuzumab/Paclitaxel		*ATRX*, *BTG2*, *IGHMBP2*, *MUTYH*, *PNKP*, *PPP1R15A*, *SEM4A4*, *P73*, *XPC*	*BRCA1*, *EXO1*, *FEN1*, *MSH2*, *NBN*, *OGG1*, *PRKDC*, *RAD18*, *RAD21*, *XRCC2*
[^212^Pb]-Trastuzumab /Gemcitabine		*BTG2*, *ERCC1*, *SEMA4A*, *P73*, *XPC*	*BRCA1*, *DMC1*, *EXO1*, *FANCG*, *FEN1*, *MSH2*, *MSH3*, *NBN*, *NTHL1*, *PRKDC*, *RAD18*, *RAD21*, *RAD51B*, *UNG*, *XRCC2*

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
