# Peer review of "Gene Abnormalities and Modulated Gene Expression Associated with Radionuclide Treatment: Towards Predictive Biomarkers of Response"

_genes, 2024, doi:10.3390/genes15060688_

Round 1
Reviewer 1 Report
Comments and Suggestions for Authors It’s a well-written review to summarize different studies to show how gene expression changes associated with responses for patients with targeted radiotherapy. My comments to improve the manuscript:Major:
- In table 1, why not list the number of patients for all studies?
- In line 203-204, it was mentioned that small number of patinets were determined hinder any definitive conclusions. However, Kidd et al [32] was having even less patients. was the conclusion from kidd et al [32] also not definitive? It might worth discussing a bit about the number of patinets also affects the results in the discussion part.
- two table 3 and one of them spelled table as tale.
- Were line 216 and line 262 subsection title?
Author Response
- Thank you for your helpful suggestions
- Number of patients used in analysis now stated in table 1.
- Low patient numbers in some of the clinical studies is certainly an issue which was briefly covered in paragraph 5 of the Discussion. I have added a sentence to further emphasise this flaw in some studies.
Reviewer 2 Report
Comments and Suggestions for Authors
The manuscript is well-written and well-organized. However, I have a few comments for your reference that you may find helpful. Firstly, it would be beneficial to add a subtitle number to facilitate easier tracking. Secondly, it would be better if you could include two figures to demonstrate the history and application of MRT.
Author Response
Thank you for your helpful suggestions
- Sections and subsections now numbered.
- A figure that combined significant historical and application dates for MRT now included (in a single figure – I trust that this is acceptable).